# Coumarin’s Anti-Quorum Sensing Activity Can Be Enhanced When Combined with Other Plant-Derived Small Molecules

**DOI:** 10.3390/molecules26010208

**Published:** 2021-01-03

**Authors:** Dmitry Deryabin, Kseniya Inchagova, Elena Rusakova, Galimzhan Duskaev

**Affiliations:** Federal Research Centre of Biological Systems and Agro-technologies of the Russian Academy of Sciences, Orenburg 460000, Russia; dgderyabin@yandex.ru (D.D.); ksenia.inchagova@mail.ru (K.I.); gduskaev@mail.ru (G.D.)

**Keywords:** coumarins, quorum sensing, QS inhibitors, plant-derived molecules, *Chromobacterium violaceum*

## Abstract

Coumarins are class of natural aromatic compounds based on benzopyrones (2H-1-benzopyran-2-ones). They are identified as secondary metabolites in about 150 different plant species. The ability of coumarins to inhibit cell-to-cell communication in bacterial communities (quorum sensing; QS) has been previously described. Coumarin and its derivatives in plant extracts are often found together with other small molecules that show anti-QS properties too. The aim of this study was to find the most effective combinations of coumarins and small plant-derived molecules identified in various plants extracts that inhibit QS in *Chromobacterium violaceum* ATCC 31532 violacein production bioassay. The coumarin and its derivatives: 7-hydroxycoumarin, 7.8-dihydroxy-4-methylcoumarin, were included in the study. Combinations of coumarins with gamma-octalactone, 4-hexyl-1.3-benzenediol, 3.4.5-trimethoxyphenol and vanillin, previously identified in oak bark (*Quercus cortex*), and eucalyptus leaves (*Eucalyptus viminalis*) extracts, were analyzed in a bioassay. When testing two-component compositions, it was shown that 7.8-dihydroxy-4-methylcoumarin, 4-hexyl-1.3-benzendiol, and gamma-octalactone showed a supra-additive anti-QS effect. Combinations of all three molecules resulted in a three- to five-fold reduction in the concentration of each compound needed to achieve EC_50_ (half maximal effective concentration) against QS in *C. violaceum* ATCC 31532.

## 1. Introduction

Coumarins are a class of natural compounds based on benzopyrones (2H-1-benzopyran-2-ones) [1]. These compounds can be classified depending on the core’s structure and the presence of substituents. There are the “simplest” coumarins (e.g., coumarin and dihydrocoumarin), followed by oxy-, meth-oxy-, and methylenedioxycoumarins with various substitutions in benzene/pyron rings (e.g., umbelliferon, 3-hydroxycoumarin, and scopoletin). The furancoumarins (e.g., bergamotin) contain an additional condensed furan core. Other, more structurally complex compounds are the result of coumarin condensation with pyran, benzene, and benzofuran rings. Most of the compounds of this class in plants are found in the free state, and only a small number are found in glycosides with D-glucose attached to the C6, C7, or C8 atoms of the coumarin nucleus [2].

Currently, coumarins are identified as secondary metabolites in about 150 different plant species distributed in almost 30 families, of which the most important are *Rutaceae, Umbelliferae, Clusiaceae, Guttiferae, Caprifoliaceae, Oleaceae, Nyctaginaceae,* and *Apiaceae* [3]. These substances are synthesized from phenylalanine via the shikimic acid formation pathway (hydroxylation, glycolysis, and cyclization of cinnamic acid) [4], and often, several different coumarins are found in the same plant.

Coumarins proposed for medical use due to their proven biological activity. They are showed anti-ulcerogenic [5], antiparasitic [6], anti-inflammatory [7,8,9,10,11], and other properties [12,13,14]. They are also antioxidant [15], and anticoagulant compounds [16,17,18,19]. As such, they can be defined as new pharmaceutical candidates [20].

The ability of coumarins to inhibit cell-to-cell communication in bacterial communities—better known as “quorum sensing” (QS)—has been discovered relatively recently. Briefly: QS is a special type of regulator of bacterial gene expression that functions at a high microbial population density. Depending on the chemical nature of the autoinducer, QS can be divided into several types: 1) LuxI/LuxR type (autoinducers–acylated homoserin lactones); 2) type II QS systems (autoinducers–furanone derivatives); 3) QS systems with Gram-positive bacteria (autoinducers–short oligopeptides); 4) QS systems with autoinducers of various natures (e.g., epinephrine, norepinephrine). The first of the described and most common QS systems is a two-component system of the LuxI/LuxR type inherent in many bacterial pathogens [21] where it activates the synthesis of virulence factors and the biofilms formation.

Because the search for plant-derived molecules with anti-QS activity is very actual, the coumarins are interesting object for this screening. Experimental observations of the anti-QS activity of coumarin are mainly related to Gram-negative bacteria that use a LuxI/LuxR type communication system, e.g., *Pseudomonas. aeruginosa* (in which coumarin suppress of phenazine biosynthesis, and motility) and *Aliivibrio fischeri* (coumarin inhibits the bioluminescence) [22]. Another simple coumarin, i.e., dihydrocoumarin, effectively inhibited QS-dependent biosynthesis of violacein in *Chromobacterium violaceum* [23]. The subsequent comparative analysis of seven hydroxycoumarin derivatives in relation to the violacein biosynthesis in *C. violaceum* showed that the promising anti-QS effect is characteristic of 3-hydroxycoumarin [24]. The identification of other functional substitutions of coumarin core, which led to disruption of the bacterial biofilm formation and, at the same time, inhibits QS development, was done by Reen [4]. This variant of bioactivity was also characteristic of a larger group of compounds, including furanocumarins: bergamottin and 6.7-dihydroxybergamottin. Interestingly, the last two compounds found in citrus fruits (*Citrus bergamia, Citrus maxima*, and *Citrus × paradisi*) showed their activity against bacteria that use both LuxI/LuxR-type and type II QS system [25].

Significantly, coumarin and its derivatives are often found together with other plant-derived small molecules that also have anti-QS properties. At the same time, our previous studies have shown that such molecules can act synergistically in a single plant [26,27]; however, until now, the possible combination of coumarins and small plant-derived molecules that are part of various plant extracts remains open.

The aim of this study was to find the most effective combinations of coumarins with small plant-derived molecules previously identified in extracts of oak bark (*Quercus cortex)*, and eucalyptus leaves (*Eucalyptus viminalis*) to inhibit LuxI/LuxR-type quorum sensing in *C. violaceum* ATCC (American Type Culture Collection) 31532.

## 2. Results

### 2.1. Effect of Coumarin and Its Derivatives in Chromobacterium violaceum ATCC 31532 Violacein Production Bioassay

Cultivation of *C. violaceum* ATCC 31532 with coumarin, 7-hydroxycoumarin, and 7.8-dihydroxy-4-methylcoumarin followed registration of the optical density values (OP_450_) of bacterial biomass and violacein production (OP_600_), allowed us to evaluate their effect on the growth and QS-dependent biosynthesis in the bioassay. All tested components showed antibacterial activity as follows: minimum inhibitory concentration (MIC_50_) = 2.689 mg/mL and MIC_100_ = 3.650 mg/mL for coumarin; MIC_50_ = 0.497 mg/mL and MIC_100_ = 1.267 mg/mL for 7-hydroxycoumarin and MIC_50_ = 0.325 mg/mL and MIC_100_ = 2.400 mg/mL for 7.8-dihydroxy-4-methylcoumarin. Simultaneously, sub-inhibitory concentrations of these compounds provided an anti-QS effect evaluated by inhibition on pigment violacein production, which was expressed as follows: effective concentration (EC_50_) = 1.105 mg/mL and EC_100_ = 3.650 mg/mL for coumarin; EC_50_ = 0.199 mg/mL and EC_100_ = 0.633 mg/mL for 7-hydroxycoumarin and EC_50_ = 0.150 mg/mL and EC_100_ = 1.200 mg/mL for 7.8-dihydroxy-4-methylcoumarin (Table 1). Thus, the highest anti-QS activity was demonstrated by 7.8-dihydroxy-4-methylcoumarin, and this coumarin derivative was taken for further studies on the combination of small plant-derived molecules.

### 2.2. Analysis of the Combined Use of Coumarin Derivatives (7.8-Dihydroxy-4-methylcumarin) with Other Small Plant-Derived Molecules in C. violaceum ATCC 31532

Results were obtained for combinations of 7.8-dihydroxy-4-methylcumarin (identified in *Baikal skullcap* extract) with small plant-derived molecules from oak bark (4-hexyl-1.3-benzenediol, 3.4.5-trimethoxyphenol, vanillin) or in eucalyptus leaves (gamma-octalactone), which have own anti-QS activity preliminary in *C. violaceum* ATCC 3153 bioassay. While most coumarins combinations showed simple additivity, some two-component mixtures led to pronounced mutual strengthening of anti-QS activity, which was evaluated as a synergetic (supra-additive) effect in 2D isobolographic analysis. The supra-additivity was revealed in combination of 7.8-dihydroxy-4-methylcoumarin with 4-hexyl-1.3-benzenediol (Figure 1a) as well as in 7.8-dihydroxy-4-methylcoumarin and gamma-octalactone mixtures (Figure 1b), where cultivation of *C. violaceum* ATCC 31532 in media enriched these paired molecular compositions showed a two-to four-fold decrease in the concentrations of each compound to achieve 50% inhibition of QS-controlled violacein biosynthesis. In turn, some combination of small molecules from oak bark and eucalyptus leaves showed the antagonistic (infra-additive) effect achieved in the compositions gamma-octalactone and vanillin, gamma-octalactone and 3.4.5-trimethoxyphenol, while the combinations gamma-octalactone and 4-hexyl-1.3-benzenediol have a supra-additive effect.

On this basis the further research on the combined use of the small molecules from Baikal skullcap, oak bark, and eucalyptus leaves included 7.8-dihydroxy-4-methylcoumarin, 4-hexyl-1.3-benzenediol, and gamma-octalactone, because all pairwise combination of these compounds showed supra-additive anti-QS effect in *C. violaceum* ATCC 31532 bioassay.

### 2.3. Evaluation of the Effect of a Three-Component Composition of Small Plant-Derived Molecules on the Quorum Sensing in C. violaceum ATCC 31532

The bioassay of small plant-derived molecules composition, which included various ratios of 7.8-dihydroxy-4-methylcoumarin, 4-hexyl-1.3-benzenediol and gamma-octalactone, confirmed supra-additive effect of two-component composition, and first showed synergy of three-component composition which manifested in the location of majority of the experimental points below the 3D isobole plane (Figure 2).

Figure 2 shows a 3D isobole in the form of a triangle, the vertices of which connect the concentrations of each compounds that cause the same biological effect EC_50_ (50% inhibition of violacein biosynthesis in *C. violaceum* ATCC 31532 bioassay). At the point of maximum supra-additive effect, with the ratio of tested compounds set to 0.6:1:0.8, the concentrations of each compound were three- to five- lower than the concentrations of individual compounds required to achieve EC_50_. Importantly, the supra-additive effect was detected in at least 85% of samples with various rations of 7.8-dihydroxy-4-methylcoumarin, 4-hexyl-1.3-benzenediol and gamma-octalactone. 

Thus, the results of our study described original compositions of various in structure small plant-derived molecules of different origins: 7.8-dihydroxy-4-methylcoumarin from Baikal skullcap, 4-hexyl-1.3-benzenediol from oak bark, and gamma-octalactone from eucalyptus leaves, which enhance each other’s anti-quorum activity. In such composition, the content of coumarin’s derivative can be significantly decreased while maintaining the anti-QS effect, which makes it possible to avoid unfavorable manifestations of the bioactivity of this group of compounds.

## 3. Discussion

Coumarins have a wide range of biological properties including antiviral, antimicrobial, anti-inflammatory, and other bioactivities. Some coumarins are approved for use in the treatment of various diseases [28,29,30,31]. The most important are vitamin K antagonists, such as warfarin, phenprocumone, or acenocumarol, which are used as anticoagulants [32,33]. Numerous studies have also shown that these compounds do not exhibit significant toxicity to humans and animals (LD_50_ = 275 mg/kg), and are only moderately toxic to the liver and kidneys [34]. In this study we used coumarin and its derivatives at concentrations significantly lower than their LD_50_ for mammals [35,36,37].

The novel variant of coumarins bioactivity is anti-QS effect that disrupt cell-to-cell chemical communication in bacteria. In this study we continued this direction and followed the path of analyzing the coumarins compositions with other plant-derived molecules in order to enhance the anti-QS effect.

Using *C. violaceum* ATCC 31532 bioassay we found anti-QS effect at sub-inhibitory concentrations of coumarin, 7-hydroxycoumarin, and 7.8-dihydroxy-4-methylcoumarin, that is in good agreement with the same activity of other coumarin derivatives: esculetin (6.7-dihydroxycoumarin) [38,39], scopoletin (7-hydroxy-5-methoxycoumarin) [40], furanocoumarin [25], nodakenetin, fraxin [41], and fizetin [42]. This allows us to state the universality of this bioactivity variant for compounds in this group.

Important, that 7.8-dihydroxy-4-methylcoumarin was characterized as the most effective anti-QS compound in this study. This data has not been previously reported anywhere, whereas the described 7.8-dihydroxy-4-methylcoumarin bioactivity comprised its antioxidant properties only [43,44]. At the same time, its structural features, particularly the hydroxy groups positions, well-corresponded to all known anti-QS active coumarins [23,24,25]. Our results were consistent with those of Yang et al. which noted a significant increase in the antibacterial effect upon the hydroxylation of coumarins at positions 6, 7, or 8 [45]. Our data also partially agreed with the studies of Lee et al., who showed that hydroxylation at position 7 increased anti-QS activity, while dihydroxylation of coumarin at positions 6 and 7 decreased this activity in comparison to conventional coumarin [46].

The next step was to combine 7.8-dihydroxy-4-methylcoumarin with other small plant-derived molecules in order to get mutual potentiation of the final anti-QS effect. The originality of the proposed approach was that we combined molecules from different plant sources. When testing two-component compositions, it was shown for the first time that 7.8-dihydroxy-4-methylcoumarin, 4-hexyl-1.3-benzendiol, and gamma-octalactone demonstrated a synergetic (supra-additive) anti-Qs effect, and combining all three molecules together decreased the concentration of each compound required to achieve EC_50_ in the composition by three-to-five-fold.

Discussing the mechanism of revealed super-additive effect we assumed that it based on the complementary bioactivity mechanisms for each compound (Figure 3). In this concept, gamma-octolactone is structurally close to LuxI/LuxR quorum sensing autoinducers (acylated homoserine lactones) and probably interferes with him for receptor binding. The 4-hexyl-1.3-benzenediol have a not fully identified mechanism, shown in one of our previous study [47,48], which repress the sensitivity of bacterial cells to autoinducers. Coumarins are characterized by a special mechanism through inhibition of the metabolism of cyclic 3’,5’-diguanilate (c-di-GMP), an intracellular intermediate that is involved in the regulation of bacterial exopolysaccharide synthesis, biofilm formation, adhesion, and virulence [24]. Doing together, these three compounds block the quorum sensing development at different stages, which is manifested in their super-additive anti-QS effect (Figure 3).

The practical aspect of these results assumes the combined use of coumarin derivatives and other small plant-derived molecules to combat bacterial pathogens of plants, animals, and humans that use quorum sensing systems for the induction of virulence factors and biofilm formation. The implementation of this approach is to use an artificial molecular composition consists of 7.8-dihydroxy-4-methylcoumarin, 4-hexyl-1.3-benzendiol, and gamma-octalactone or the plant materials mixtures with high content of these compounds: Baikal skullcap (*Scutellaria baicalensis*), oak bark (*Quercus cortex*), and eucalyptus leaves (*Eucalyptus viminalis*). Due to the high biological activity of these compositions, they can become a substitute for antibiotics in feeding of farm animals, and should also be considered as candidate pharmaceuticals for further preclinical and clinical studies.

## 4. Materials and Methods

### 4.1. Chemical Compounds

Coumarin and its derivatives were used to inhibit QS in *Chromobacterium violaceum* ATCC 31532: coumarin (2H-chromene-2OH; CAS 91-64-5) (Figure 4A), 7-hydroxycoumarin (7-hydroxy-2H-1-benzopyran-2-one; CAS 93-35-6) (Figure 4B), and 7.8-dihydroxy-4-methylcoumarin (4-methyldafnetin; CAS 2107-77-9) (Figure 4C).

Small plant-derived molecules with previously reported anti-QS activity that were identified in extracts of oak bark (*Quercus cortex*), and eucalyptus leaves (*Eucalyptus viminalis*), were tested in combinations with coumarin derivatives. The analysis included gamma-octalactone (2(3H)-furanone; CAS 147852-83-3), 4-hexyl-1.3-benzenediol (4-n-propylresorcinol; CAS 13331-19-6), 3.4.5-trimethoxyphenol (antiarol; CAS 642-71-7), vanillin (4-hydroxy-3-methoxy benzaldehyde; CAS 121-33-5).

Each of these compounds had a purity at least 99% and was purchased from Sigma-Aldrich (St. Louis, MO, USA).

### 4.2. Bacterial Strain

The wild strain of *C. violaceum* ATCC 31532 that possessed a two-component LuxI/LuxR-type QS system, was used in bioassay. In this strain CviI synthase (LuxI analog) produce autoinducer N-hexanoyl-L-homoserin lactone (C6-AHL) which bond CviR receptor protein (LuxR analog) and activate QS-controlled transcription of several target genes including *vioABEDC* operon [49]. The encoded VioA, VioB, VioE, VioD, and VioC proteins form a biosynthetic pathway for blue-violet pigment violacein with a maximum absorption at 585 nm. The amount of pigment in the bacterial culture allowed us to directly assess the QS activity.

### 4.3. Methods for Investigating Anti-QS Activity of Coumarin Derivatives in C. violaceum ATCC 31532 Bioassay

To determine the anti-QS activity of each compound in Luria-Bertani (LB) broth, double dilutions (n × 2) were prepared. The similar samples of LB-broth that did not contain tested compounds were used as positive (growth of the test strain) and negative (sterile) controls. Glass vessels containing 2 mL of experimental dilutions or control samples were inoculated 20 µl of a one-day *C. violaceum* ATCC 31532 culture and cultivated in a static mode at 27 °C. The results were evaluated using a multifunctional microplate reader Infinite 200 PRO (Tecan, Männedorf, Switzerland). The optical density at 450 ± 5 nm (OP_450_) measured the bacterial biomass and evaluated the effect of the studied compounds on bacterial growth, while the violacein pigment after its ethanol extraction was determined at 600 ± 5 nm (OP_600_), which was an indicator of the effect on the QS system. The absorption values of the negative control were subtracted. The antibacterial effect of the studied compounds was presented by the MIC_100_ and MIC_50_ values, which were minimal inhibitory concentrations that caused 100% and 50% growth suppression for the test strain relative to the positive control. The inhibition of quorum sensing was expressed as EC_100_ and EC_50_ values, which were equal to 100% and 50% inhibition of violacein pigment biosynthesis in grown culture, respectively.

### 4.4. Evaluation of the Combined Use of Coumarins and Small Plant-Derived Molecules against in C. violaceum ATCC 31532

To examine the combined effect, double dilutions of test compounds were introduced into plastic 96-well plates in perpendicular directions (connection X: connection Y), so that each well contained their individual ratio. Comparison samples were a series of dilutions containing only one of the tested compounds, as well as positive and negative controls. Further inoculation of *C. violaceum* ATCC 31532, cultivation, and recording of the study results were performed as described above. The effect of the paired compositions was evaluated using isobolographic analysis [49], which is based on the construction of 2D isoboles (i.e., lines connecting the EC_50_ values for the studied compounds X and Y, on the abscissa and ordinate axes) followed by drawing points on this graph which corresponding to the combined effect of compounds X and Y at different concentration ratios. The location of such points on the isobole line corresponded to an additivity (summation effect), their placement above the isobole line described an infra-additive effect (antagonism), and below the isobole showed a supra-additive (synergetic) effect.

Studies of a three-component composition were beginning from the formation of a series of 96-well plates containing the double dilutions of compounds X and Y, as described above. On the next step the wells in each plate were filled with a certain concentration of compound Z (i.e., the total number of used 96-well plates was equal to the number of tested concentrations of compound Z). The control samples were a dilution series containing only compound X, only compound Y, or only compound Z, as well as positive and negative controls. Wells were inoculated with *C. violaceum* ATCC 31532, cultivated and analyzed as described above. The effect of the three-component compositions was evaluated using three-dimensional (3D) isoboles plotted based on the EC_50_ values for each compound.

### 4.5. Statistical Analysis

All values presented a mean of the 5 experiments. The obtained results were processed using methods of statistical variance in Excel for Windows 10.

## 5. Conclusions

In this study, coumarin and its derivatives were tested against quorum sensing in *Chromobacterium violaceum* ATCC 31532, and promising activity was shown in 7.8-dihydroxy-4-methylcoumarin. This compound previously detected in Baikal skullcap (*Scutellaria baicalensis*) was combined with other small plant-derived molecules identified in extracts of oak bark (*Quericus cortex*), and eucalyptus leaves (*Eucalyptus viminalis*). It has been shown that 7.8-dihydroxy-4-methylcoumarin, 4-hexyl-1.3-benzenediol and gamma-octalactone exhibit a supra-additive anti-QS effect in two-component combinations, and the combination of all three molecules reduces the concentration of each of them required for reaching EC_50_ against QS, by three- to five- times. It was proposed that the super-additive effect is based on various bioactivity mechanisms of tested molecules, which disrupt the QS development at different stages. The results provide a use for small plant-derived molecule compositions plant materials in the feeding of farm animals, replacing the similar use of prohibited feed antibiotics [50], and also determines the prospects for their testing against human pathogens that use QS to induce virulence factors and biofilm development.

## Figures and Tables

**Figure 1 molecules-26-00208-f001:**
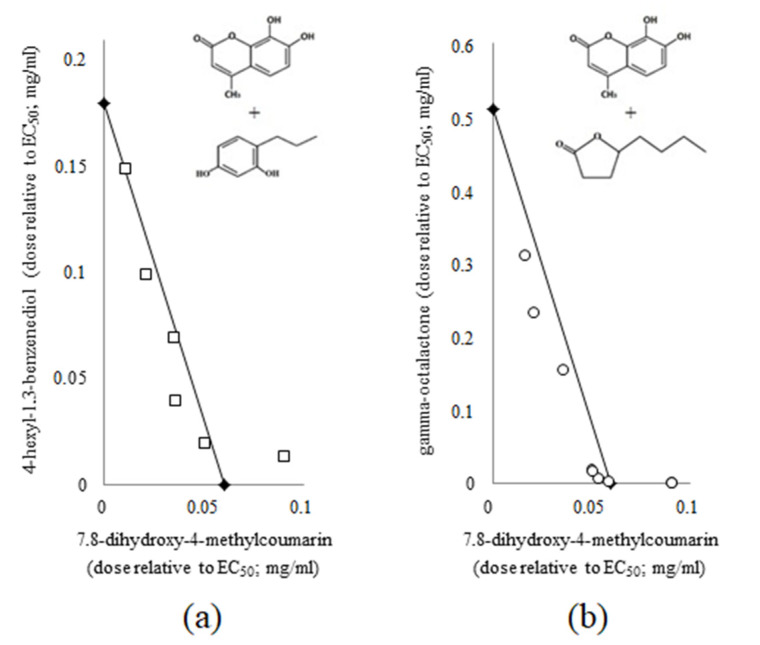
2D isobolographic analysis of the combined use of 7.8-dihydroxy-4-methylcoumarin and 4-hexyl-1.3-benzenediol (**a**), 7.8-dihydroxy-4-methylcoumarin and gamma-octalactone (**b**) on the QS-controlled violacein biosynthesis in *C. violaceum* ATCC 31532. Isoboles are represented as straight lines connecting the EC_50_ concentrations of each compounds. The points under the isoboles correspond to the supra-additive effect.

**Figure 2 molecules-26-00208-f002:**
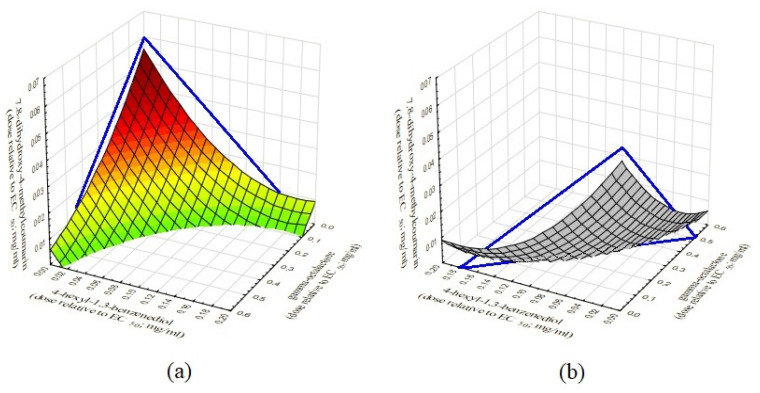
3D isobolographic analysis of the combined use of 7.8-dihydroxy-4-methylcoumarin, 4-hexyl-1.3-benzenediol, and gamma-octalactone against the QS-controlled violacein biosynthesis in *C. violaceum* ATCC 31532: (**a**) front view; (**b**) back view. The 3D isobole is represented as a blue triangle, the vertices of which correspond to 50% violacein biosynthesis inhibition (EC_50_) for each compound; the “sail” plane show the supra-additive effect of the experimental samples under the 3D isobole plane.

**Figure 3 molecules-26-00208-f003:**
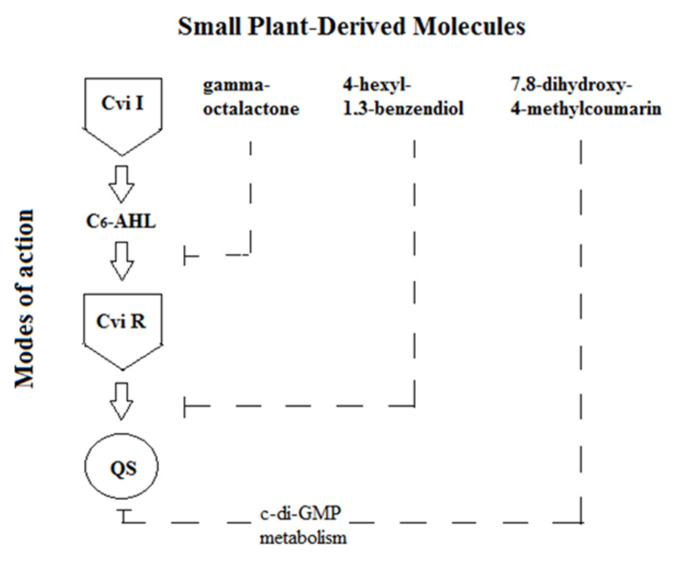
Proposed mechanism of supra-additive anti-QS effect of molecular composition consists of 7.8-dihydroxy-4-methylcoumarin, 4-hexyl-1.3-benzendiol, and gamma-octalactone.

**Figure 4 molecules-26-00208-f004:**
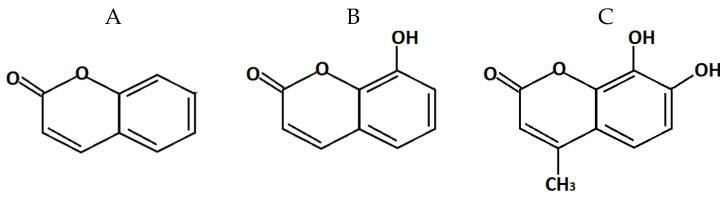
Structural formula of coumarin (**A**), 7-hydroxycoumarin (**B**), and 7.8-dihydroxy-4-methylcoumarin (**C**).

**Table 1 molecules-26-00208-t001:** Effects of coumarin, 7-hydroxycoumarin and 7.8-dihydroxy-4-methylcoumarin (mg/mL) on growth and QS-controlled violacein pigment biosynthesis in *C. violaceum* ATCC 31532.

Tested Compound	Characteristics of Antibacterial Activity, mg/mL		Characteristics of Anti-QS Activity, mg/mL	
	MIC_100_	MIC_50_	EC_100_	EC_50_
Coumarin	3.650	2.689	3.650	1.105
7-hydroxycoumarin	1.267	0.497	0.633	0.199
7.8-dihydroxy-4-methylcoumarin	2.400	0.325	1.200	0.150

## Data Availability

The data that support the findings of this study are available from the corresponding author, upon reasonable request.

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
