# Peer review of "Coumarin’s Anti-Quorum Sensing Activity Can Be Enhanced When Combined with Other Plant-Derived Small Molecules"

_molecules, 2021, doi:10.3390/molecules26010208_

Round 1
Reviewer 1 Report
As far as I am concerned the manuscript is well written. The subject area of research is significant in knowledge development. The introduction is interesting and correct. Test methods selected correctly.
The results of the research were presented very well. The manuscript used the latest literature on the subject.
I have only one editorial comments:
L.40 unnecessary space:oxy -,
l. 167 necessary space before [20,21]
Author Response
Thank you for your kind review. Your recommended corrections have been made to the revised manuscript.
L.40 unnecessary space:oxy -, edited
L.167 necessary space before [20,21] – edited.
Reviewer 2 Report
This manuscript presents the use of combinations of coumarin molecules on the quorum sensing (QS) inhibition. Combinations of specific coumarin molecules were identified as having the highest inhibitory action, which were for combinations not found in nature. The results are interesting and this study should appeal to a broad audience of scientists working on natural products and synthesis of new pharmaceutical drugs based on natural products. I have minor comments that need to be addressed in order to make the article more accessible to readers and improve the conclusions. Once these are addressed, no further comments are required. Comments:
1. Authors start counting their figures from Figure 2. Figure 1 appears in section 4, Materials and Methods. If this section is to be kept at the end of the manuscript, Figure 1 should still be moved and cited before Figure 2.
2. Also, in Figure 2, represent decimals by "." instead of commas, as in 0.1 instead of 0,1. Include the axis ticks for the scales.
3. In lines 181 and 182, authors state "It is only known that 7,8-dihydroxy-4-methylcoumarin (7,8-DHMC) is capable of 181 powerful purification from free radicals." This statement is unclear. Is the compound purified by free radicals, or does it remove free radicals?
4. Lines 209-211 page 6: Authors state "The results demonstrate formation of an original "nature-like" composition of small molecules of plant origin that do not occur in the composition of a single medicinal plant but exhibit a mutually reinforcing super-additive effect against quorum sensing." What do they mean by "nature-like" when this combination is not found in nature?
5. Will there be a Conclusions section?
Author Response
Point 1: 1. Authors start counting their figures from Figure 2. Figure 1 appears in section 4, Materials and Methods. If this section is to be kept at the end of the manuscript, Figure 1 should still be moved and cited before Figure 2.
The authors thank the reviewer for a detailed review of the text and comments made.
Response 1: The numbering of the figures has been changed for their consecutive mention in the manuscript.
Point 2: Also, in Figure 2, represent decimals by "." instead of commas, as in 0.1 instead of 0,1. Include the axis ticks for the scales.
Response 2: The changes in Figure 2 are made as recommended.
Point 3: In lines 181 and 182, authors state "It is only known that 7,8-dihydroxy-4-methylcoumarin (7,8-DHMC) is capable of 181 powerful purification from free radicals." This statement is unclear. Is the compound purified by free radicals, or does it remove free radicals?
Response 3: The sentance edited as follows “The previously described 7,8-dihydroxy-4-methylcoumarin (7,8-DHMC) bioactivity was its antioxidant properties only”
Point 4: Lines 209-211 page 6: Authors state "The results demonstrate formation of an original "nature-like" composition of small molecules of plant origin that do not occur in the composition of a single medicinal plant but exhibit a mutually reinforcing super-additive effect against quorum sensing." What do they mean by "nature-like" when this combination is not found in nature?
Response 4: The sentance edited as follows "The results demonstrate formation of an original composition of small molecules of plant origin that do not occur in the composition of a single medicinal plant but exhibit a mutually reinforcing super-additive effect against quorum sensing."
Point 5: Will there be a Conclusions section?
Response 5: Conclusions section added. “Thus, the study of coumarin and its derivatives in combination with small molecules of plant origin demonstrates the prospect of using an original "nature-like" composition of small molecules of plant origin with a different mechanism of action on QS (7,8-dihydroxy-4-methylcoumarin : gamma-octalactone : 4-hexyl-1,3-benzenediol). the planned use of the results of the study provides for the inclusion of small molecule compositions in the feeding systems of farm animals, replacing the similar use of prohibited feed antibiotics [51]. At the same time, the prospective possible clinical significance of our data also implies their approbation in systems for the prevention and treatment of human infectious diseases, the pathogens of which use the quorum sensing system to induce their pathogenic potential”.
Reviewer 3 Report
The manuscript is interesting,
I have some question regarding these paper:
I suggest the authors add these recent manuscripts concerning the biological properties of coumarins in the introduction.
Küpeli Akkol E, Genç Y, Karpuz B, Sobarzo-Sánchez E, Capasso R. Coumarins and
Coumarin-Related Compounds in Pharmacotherapy of Cancer. Cancers (Basel). 2020
Jul 19;12(7):1959.
Koga H, Negishi M, Kinoshita M, Fujii S, Mori S, Ishigami-Yuasa M, Kawachi E,
Kagechika H, Tanatani A. Development of Androgen-Antagonistic Coumarinamides
with a Unique Aromatic Folded Pharmacophore. Int J Mol Sci. 2020 Aug
4;21(15):5584.
Shahzadi I, Ali Z, Baek SH, Mirza B, Ahn KS. Assessment of the Antitumor
Potential of Umbelliprenin, a Naturally Occurring Sesquiterpene Coumarin.
Biomedicines. 2020 May 18;8(5):126.
Cruz LF, Figueiredo GF, Pedro LP, Amorin YM, Andrade JT, Passos TF, Rodrigues
FF, Souza ILA, Gonçalves TPR, Dos Santos Lima LAR, Ferreira JMS, Araújo MGF.
Umbelliferone (7-hydroxycoumarin): A non-toxic antidiarrheal and antiulcerogenic
coumarin. Biomed Pharmacother. 2020 Sep;129:110432.
Takomthong P, Waiwut P, Yenjai C, Sripanidkulchai B, Reubroycharoen P, Lai R,
Kamau P, Boonyarat C. Structure-Activity Analysis and Molecular Docking Studies
of Coumarins from Toddalia asiatica as Multifunctional Agents for
Alzheimer's Disease. Biomedicines. 2020 May 2;8(5):107
Janus Ł, Radwan-Pragłowska J, Piątkowski M, Bogdał D. Coumarin-Modified CQDs
for Biomedical Applications-Two-Step Synthesis and Characterization. Int J Mol
Sci. 2020 Oct 29;21(21):8073.
Lee EJ, Kang MK, Kim YH, Kim DY, Oh H, Kim SI, Oh SY, Na W, Kang YH. Coumarin
Ameliorates Impaired Bone Turnover by Inhibiting the Formation of Advanced
Glycation End Products in Diabetic Osteoblasts and Osteoclasts. Biomolecules.
2020 Jul 15;10(7):1052.
Starzak K, Świergosz T, Matwijczuk A, Creaven B, Podleśny J, Karcz D. Anti-
Hypochlorite, Antioxidant, and Catalytic Activity of Three Polyphenol-Rich
Super-Foods Investigated with the Use of Coumarin-Based Sensors. Biomolecules.
2020 May 6;10(5):723
Usman H, Ullah MA, Jan H, Siddiquah A, Drouet S, Anjum S, Giglioli-Guviarc'h
N, Hano C, Abbasi BH. Interactive Effects of Wide-Spectrum Monochromatic Lights
on Phytochemical Production, Antioxidant and Biological Activities of Solanum
xanthocarpum Callus Cultures. Molecules. 2020 May 8;25(9):2201
Nasser MI, Zhu S, Hu H, Huang H, Guo M, Zhu P. Effects of imperatorin in the
cardiovascular system and cancer. Biomed Pharmacother. 2019 Dec;120:109401.
Duan J, Shi J, Ma X, Xuan Y, Li P, Wang H, Fan Y, Gong H, Wang L, Pang Y,
Pang S, Yan Y. Esculetin inhibits proliferation, migration, and invasion of
clear cell renal cell carcinoma cells. Biomed Pharmacother. 2020 May;125:110031
Olanlokun JO, Bodede O, Prinsloo G, Olorunsogo OO. Comparative antimalarial,
toxicity and mito-protective effects of Diospyros mespiliformis Hochst. ex A.
- and Mondia whitei (Hook.f.) Skeels on Plasmodium berghei infection in mice.
J Ethnopharmacol. 2020 Nov 12:113585.
Bihani T. Plumeria rubra L.- A review on its ethnopharmacological,
morphological, phytochemical, pharmacological and toxicological studies. J
Ethnopharmacol. 2021 Jan 10;264:113291.
Urbagarova BM, Shults EE, Taraskin VV, Radnaeva LD, Petrova TN, Rybalova TV,
Frolova TS, Pokrovskii AG, Ganbaatar J. Chromones and coumarins from
Saposhnikovia divaricata (Turcz.) Schischk. Growing in Buryatia and Mongolia and
their cytotoxicity. J Ethnopharmacol. 2020 Oct 28;261:112517.
Williams KJ, Gieling RG. Preclinical Evaluation of Ureidosulfamate Carbonic
Anhydrase IX/XII Inhibitors in the Treatment of Cancers. Int J Mol Sci. 2019 Dec
2;20(23):6080.
Are the doses used toxic?
The manuscript would benefit from inclusion of introducing/bridging sentences between the individual parts of the "Results" that explain the logical order and rationale for the experiments
In the conclusions , the Authors should highlight the possible clinical significance of their findings
Author Response
Point 1: 1. I suggest the authors add these recent manuscripts concerning the biological properties of coumarins in the introduction.
Küpeli Akkol E, Genç Y, Karpuz B, Sobarzo-Sánchez E, Capasso R. Coumarins and
Coumarin-Related Compounds in Pharmacotherapy of Cancer. Cancers (Basel). 2020
Jul 19;12(7):1959.
Koga H, Negishi M, Kinoshita M, Fujii S, Mori S, Ishigami-Yuasa M, Kawachi E,
Kagechika H, Tanatani A. Development of Androgen-Antagonistic Coumarinamides
with a Unique Aromatic Folded Pharmacophore. Int J Mol Sci. 2020 Aug
4;21(15):5584.
Shahzadi I, Ali Z, Baek SH, Mirza B, Ahn KS. Assessment of the Antitumor
Potential of Umbelliprenin, a Naturally Occurring Sesquiterpene Coumarin.
Biomedicines. 2020 May 18;8(5):126.
Cruz LF, Figueiredo GF, Pedro LP, Amorin YM, Andrade JT, Passos TF, Rodrigues
FF, Souza ILA, Gonçalves TPR, Dos Santos Lima LAR, Ferreira JMS, Araújo MGF.
Umbelliferone (7-hydroxycoumarin): A non-toxic antidiarrheal and antiulcerogenic
coumarin. Biomed Pharmacother. 2020 Sep;129:110432.
Takomthong P, Waiwut P, Yenjai C, Sripanidkulchai B, Reubroycharoen P, Lai R,
Kamau P, Boonyarat C. Structure-Activity Analysis and Molecular Docking Studies
of Coumarins from Toddalia asiatica as Multifunctional Agents for
Alzheimer's Disease. Biomedicines. 2020 May 2;8(5):107
Janus Ł, Radwan-Pragłowska J, Piątkowski M, Bogdał D. Coumarin-Modified CQDs
for Biomedical Applications-Two-Step Synthesis and Characterization. Int J Mol
Sci. 2020 Oct 29;21(21):8073.
Lee EJ, Kang MK, Kim YH, Kim DY, Oh H, Kim SI, Oh SY, Na W, Kang YH. Coumarin
Ameliorates Impaired Bone Turnover by Inhibiting the Formation of Advanced
Glycation End Products in Diabetic Osteoblasts and Osteoclasts. Biomolecules.
2020 Jul 15;10(7):1052.
Starzak K, Świergosz T, Matwijczuk A, Creaven B, Podleśny J, Karcz D. Anti-
Hypochlorite, Antioxidant, and Catalytic Activity of Three Polyphenol-Rich
Super-Foods Investigated with the Use of Coumarin-Based Sensors. Biomolecules.
2020 May 6;10(5):723
Usman H, Ullah MA, Jan H, Siddiquah A, Drouet S, Anjum S, Giglioli-Guviarc'h
N, Hano C, Abbasi BH. Interactive Effects of Wide-Spectrum Monochromatic Lights
on Phytochemical Production, Antioxidant and Biological Activities of Solanum
xanthocarpum Callus Cultures. Molecules. 2020 May 8;25(9):2201
Nasser MI, Zhu S, Hu H, Huang H, Guo M, Zhu P. Effects of imperatorin in the
cardiovascular system and cancer. Biomed Pharmacother. 2019 Dec;120:109401.
Duan J, Shi J, Ma X, Xuan Y, Li P, Wang H, Fan Y, Gong H, Wang L, Pang Y,
Pang S, Yan Y. Esculetin inhibits proliferation, migration, and invasion of
clear cell renal cell carcinoma cells. Biomed Pharmacother. 2020 May;125:110031
Olanlokun JO, Bodede O, Prinsloo G, Olorunsogo OO. Comparative antimalarial,
toxicity and mito-protective effects of Diospyros mespiliformis Hochst. ex A.
and Mondia whitei (Hook.f.) Skeels on Plasmodium berghei infection in mice.
J Ethnopharmacol. 2020 Nov 12:113585.
Bihani T. Plumeria rubra L.- A review on its ethnopharmacological,
morphological, phytochemical, pharmacological and toxicological studies. J
Ethnopharmacol. 2021 Jan 10;264:113291.
Urbagarova BM, Shults EE, Taraskin VV, Radnaeva LD, Petrova TN, Rybalova TV,
Frolova TS, Pokrovskii AG, Ganbaatar J. Chromones and coumarins from
Saposhnikovia divaricata (Turcz.) Schischk. Growing in Buryatia and Mongolia and
their cytotoxicity. J Ethnopharmacol. 2020 Oct 28;261:112517.
Williams KJ, Gieling RG. Preclinical Evaluation of Ureidosulfamate Carbonic
Anhydrase IX/XII Inhibitors in the Treatment of Cancers. Int J Mol Sci. 2019 Dec
2;20(23):6080.
The authors thank the reviewer for a detailed review of the text and comments made.
Response 1: In accordance with these recommendations, the following corrections are made in the manuscript: In the" Introduction", the following fragment is added “Coumarins are widely used for medical purposes due to their proven biological activity. They use as antidiarrheal and antiulcerogenic [Cruz], antiparasitic [Olanlokun], antitumor and anti-inflammatory agents [Williams KJ, Küpeli Akkol E, Shahzadi I, Duan J, Nasser], as well as biologically active [Usman, Bihani, Urbagarova] antioxidant [Starzak K] and anticoagulant compounds [5 Takomthong, Janus, Lee] what defined the view of them as new candidate pharmaceuticals [Koga]”.
Point 2: Are the doses used toxic?
Response 2: In the" Discussion", the following fragments are added: “In our study, we used the studied compounds at concentrations significantly lower than their LD50 for mammals [López et al, Aguedo et al, Inchagova et al]”
“In the context of the work it is fundamentally important that the combined use of these compounds in the composition allow to significantly reduce the concentration of each of these needed to ЕС50 effect on "quorum sensing", which further offset by the potential toxicity.”
Point 3: The manuscript would benefit from inclusion of introducing/bridging sentences between the individual parts of the "Results" that explain the logical order and rationale for the experiments
Response 3: When carrying out a proof reading service journal it will be done.
Point 4: In the conclusions , the Authors should highlight the possible clinical significance of their findings
Response 4: Conclusions section added. “Thus, the study of coumarin and its derivatives in combination with small molecules of plant origin demonstrates the prospect of using an original "nature-like" composition of small molecules of plant origin with a different mechanism of action on QS (7,8-dihydroxy-4-methylcoumarin : gamma-octalactone : 4-hexyl-1,3-benzenediol). the planned use of the results of the study provides for the inclusion of small molecule compositions in the feeding systems of farm animals, replacing the similar use of prohibited feed antibiotics [51]. At the same time, the prospective possible clinical significance of our data also implies their approbation in systems for the prevention and treatment of human infectious diseases, the pathogens of which use the quorum sensing system to induce their pathogenic potential”.